# Thymol@activated Carbon Nanohybrid for Low-Density Polyethylene-Based Active Packaging Films for Pork Fillets’ Shelf-Life Extension

**DOI:** 10.3390/foods12132590

**Published:** 2023-07-03

**Authors:** Aris E. Giannakas, Vassilios K. Karabagias, Dimitrios Moschovas, Areti Leontiou, Ioannis K. Karabagias, Stavros Georgopoulos, Andreas Karydis-Messinis, Konstantinos Zaharioudakis, Nikolaos Andritsos, George Kehayias, Apostolos Avgeropoulos, Charalampos Proestos, Constantinos E. Salmas

**Affiliations:** 1Department of Food Science and Technology, University of Patras, 30100 Agrinio, Greece; vkarampagias@upatras.gr (V.K.K.); aleontiu@upatras.gr (A.L.); ikaraba@upatras.gr (I.K.K.); sgeorgop@upatras.gr (S.G.); zacharioudakis.k@upatras.gr (K.Z.); nandritsos@upatras.gr (N.A.); gkechagi@upatras.gr (G.K.); 2Department of Material Science and Engineering, University of Ioannina, 45110 Ioannina, Greece; dmoschov@uoi.gr (D.M.); karydis.and@gmail.com (A.K.-M.); aavger@uoi.gr (A.A.); 3Laboratory of Food Chemistry, Department of Chemistry, National and Kapodistrian University of Athens Zografou, 15771 Athens, Greece; harpro@chem.uoa.gr

**Keywords:** activated carbon, thymol, low-density polyethylene, active packaging, control release, kinetics, pork fillets, heme iron, shelf-life

## Abstract

Τhe replacement of food packaging additives and preservatives with bio-based antioxidant/antibacterial compounds has been a common practice in recent years following the trend of bioeconomy and nanotechnology. Such bio-additives are often enclosed in nanocarriers for a controlled release process. Following this trend in this work, a thymol (TO)-rich activated carbon (AC) nanohybrid was prepared and characterized physicochemically with various techniques. This TO@AC nanohybrid, along with the pure activated carbon, was extruded with low-density polyethylene (LDPE) to develop novel active packaging films. The codenames used in this paper were LDPE/xTO@AC and LDPE/xAC for the nanohybrid and the pure activated carbon, respectively. X-ray diffractometry, Fourier-transform infrared spectroscopy, and scanning electron microscopy measurements showed high dispersity of both the TO@AC nanohybrid and the pure AC in the LDPE matrix, resulting in enhanced mechanical properties. The active film with 15 wt.% of the TO@AC nanohybrid (LDPE/15TO@AC) exhibited a 230% higher water/vapor barrier and 1928% lower oxygen permeability than the pure LDPE film. For this active film, the highest antioxidant activity referred to the DPPH assay (44.4%), the lowest thymol release rate (k_2_ ≈ 1.5 s^−1^), and the highest antibacterial activity were recorded, resulting in a 2-day extension of fresh pork fillets’ shelf-life.

## 1. Introduction

Today, the need to enhance the safety and shelf-life of foods is driving research and development (R&D) in the food industry towards the investigation of novel packaging techniques. Such techniques aim not only to protect food, but also to react with it in order to enhance the preservation and the safety of the product, and to extend its shelf-life [1]. This type of packaging is called “active food packaging” [1,2,3]. Moreover, one of the most recent trends in active food packaging is the removal of the chemical additives from the food and the active packaging, or at least to control the release of such food additives into the food [4,5]. In this spirit, it is also suggested to replace chemical food preservatives with natural, abundant compounds that have demonstrated antioxidant and antibacterial activity [4]. Such bioactive antioxidant/antibacterial compounds include essential oils (EOs) and their components, along with other phytochemicals like monoterpenes, sesquiterpenes, alcohols, phenols, aldehydes, ketones, and esters [6,7,8]. Over the past few years, considerable effort has been devoted to the development of polymers or biopolymer-based active packaging films incorporated with various EOs, such as oregano EO [9,10], thyme EO [9,10,11], basil EO [12,13], orange EO [14], lemongrass EO [15], or their derivatives, such as thymol [16], carvacrol [17], cinnamaldehyde [18], and limonene [19]. The main disadvantage of directly incorporating EOs into the polymer or biopolymer matrix is that their volatile nature leads to rapid loss and, therefore, the weakening of the active properties of the packaging. One famous proposed technique to balance this drawback is to adsorb these EOs or their derivatives into inexpensive, naturally abundant porous materials such as nanoclays, and afterwards to incorporate the resulting EO/nanoclay hybrid structure into the polymer or biopolymer matrix [20]. This approach enables the development of polymeric or biopolymer/EO/nanoclay active packaging films that exhibit enhanced mechanical, tensile, and water/oxygen barrier properties, as well as controlled release of antioxidant and antibacterial substances [21,22,23]. Nanoclays such as montmorillonite (MMT) and halloysite nanotubes (HNTs) have been extensively studied as EO nanocarriers in the development of polymeric or biopolymer/EO/nanoclay active packaging films [24,25]. Recently, well-known multifunctional materials, such as natural zeolite (NZ) [26,27,28], silica-based materials (MCM and SBA-15) [29,30], and carbon-based materials (activated carbon-AC) [31,32,33], have been suggested as nanocarriers for EOs and their derivatives because of their advanced adsorption capacity. The advantages of such materials compared to nanoclays are due to their larger specific surface areas and their ability to adsorb and load higher amounts of EOs or their derivatives. Among various carbon-based nanomaterials, the food-biomass-derived activated carbon (AC) has the advantage of low cost, non-toxicity, and degradability [31,34]. The effectiveness of AC is strongly dependent on its pore structure and surface chemistry [31]. AC is used in packaging films because of its adsorption and release capabilities. Thus, it can be used as a powerful adsorbent to remove odors and food spoilage substances from packaged foods, or as a nanocarrier for bioactive compounds to be released into food [33,35]. 

Low-density polyethylene (LDPE) is one of the most widely used materials for flexible food packaging. Although it is a polymer produced via petrochemical processes, in the future it could be produced via the bioethanol process as a bio-based material. It exhibits satisfactory strength-at-break properties up to −60 °C and good water barrier properties, but its high oxygen permeability makes it unsuitable as a food packaging material and vulnerable to oxidation deterioration phenomena of foods such as meat products. Oxidative and microbial deterioration is the most critical factor for pork meat’s degradation and the end of its shelf-life [36]. Therefore, in order to transform the bio-based LDPE into a suitable food packaging material, it is essential to modify it by enhancing its oxygen barrier properties through the development of an LDPE-based active packaging material. This will be more valuable if the modification process follows the spirit of the circular economy, which proposes the use of bio-based raw materials such as EOs and biomass-derived AC. Recently, Macca carbon (MC) powder biomass, which is derived from macadamia nuts, was incorporated into LDPE by melt-compounding and subsequent melt-extrusion processes [37]. The content of MC powder in the LDPE matrix was varied from 0.5 up to 5 wt.%. The film with MC powder at 0.5 wt.% exhibited improved characteristics of antibacterial activity and storage performance, allowing lettuce and strawberries to remain fresh outside the refrigerator for more than 7 and 5 days, respectively, but any further addition of such carbon material led to a distortion of the film. Based on this paper, our research group started to study the use of AC-based powders as novel nanoreinforcements and antimicrobial/antioxidant agent carriers, as well as the incorporation of the produced novel materials with LDPE for the development of active packaging films.

In recent studies, our group developed a “green” method for the extraction of a thymol-rich (TO) fraction from thyme oil. This oil was subsequently adsorbed on halloysite nanotubes (HNTs) and natural zeolite (NZ) to create innovative TO@HNT and TO@NZ nanohybrids [38,39]. These nanohybrids were successfully incorporated into low-density polyethylene (LDPE) to develop promising LDPE/TO@HNT and LDPE/TO@NZ active packaging films [40,41]. These active films were successful at preserving low-fat pork fillets by preventing their lipid oxidation process and heme iron loss [40,41]. In this study, we applied the same green method to incorporate TO in AC to obtain a novel TO@AC nanohybrid, with the prospect that the TO@AC nanohybrid would act as a novel TO nanocarrier by absorbing higher amounts of TO than the recently used HNT and NZ nanocarriers [40,41]. The obtained TO@AC nanohybrid was characterized physicochemically through XRD analysis and FTIR spectroscopy, as well as TG and DSC experiments. Next, the TO@AC nanohybrid was incorporated into the LDPE matrix in various contents via an extrusion-molding process to develop novel LDPE/TO@AC active packaging films. The LDPE/TO@AC film with the best packaging properties, as well as LDPE/AC and pure LDPE films, was utilized to wrap pork fillets, and their shelf-life was evaluated by measuring their lipid oxidation through the thiobarbituric-acid-reactive substances (TBARS) method, their heme iron values, and the total bacterial count values. 

The overall novelty of the present study is that, in the spirit of the circular economy, it proposes one more successful application for the valorization of a part of the 6 Mtons per year of globally landfilled spent coffee [42], combined with the production of natural preservation additives such as EOs. The final novel product will be an improved, environmentally friendly food packaging film capable of food preservation via “green methods” and the shelf-life extension of packaged products. The innovations of this work can be summarized as follows: (i) the development and characterization of novel TO@AC nanohybrids for the first time, to the best of our knowledge; (ii) the incorporation of pure AC and modified TO@AC nanohybrids into LDPE at high contents of 5, 10, and 15 wt.%; (iii) the development, characterization, and application of such novel LDPE/TO@AC active packaging films for pork fillet preservation for the first time, to the best of our knowledge; and (iv) in alignment with recent works [40,41], the identification of a direct linear relationship between the TBARS lipid oxidation values and heme iron values of pork fillets. 

## 2. Materials and Methods

### 2.1. Materials

Thyme oil used was supplied by Chemco (Via Achille Grandi, 13—13/A, 42030 Vezzano sul Crostolo, Reggio Emilia, Italy). LDPE (CAS No. 9002-88-4) with a density of 0.925 g cm^−3^ and a melting point of 116 °C was supplied by Sigma-Aldrich (Sigma-Aldrich, St. Louis, MO, USA). Fresh pork fillets of the “skalopini” type were a kind offering from Aifantis Company (Aifantis Group—Head Quarters, Acheloos Bridge, Agrinio, Greece 30100). Three fresh and deboned pork fillets of the “skalopini type” with an approximate weight of 700 g each were provided by a local meat processing plant (Aifantis Company) within one hour after the slaughter. Activated carbon (AC) from spent coffee of the students’ café at the University of Ioannina was prepared via a pyrolysis process and a treatment with KOH, as described in a previous report [42]. The Brunauer–Emmett–Teller surface area of the AC was 1372 m^2^/g, and the micropore volume was 84.6% [42]. 

### 2.2. Preparation of TO@AC Nanohybrid

The process followed for the modification of the pure AC with a TO-rich fraction was the same as the process that was recently used for the modification of HNTs and NZ [38,39]. As illustrated in Figure 1, the process was carried out in two steps: In the first step, a distillation took place using a distillation apparatus on 20 mL of pure thyme oil at 200 °C to obtain a TO-rich fraction (14 mL) and separate the most volatile limonene–cymene fraction (6 mL). In the second step, TO’s adsorption on AC occurred by placing 3 g of an AC bed above the spherical glass container containing the TO-rich fraction and evaporating the TO-rich fraction by heating it to 300 °C. Thus, the TO@AC nanohybrid was obtained and kept for further physicochemical characterization.

### 2.3. Preparation of LDPE/AC and LDPE/TO@AC Films

The preparation of LDPE/AC and LDPE/TO@AC films was performed by using a Mini Lab twin-screw extruder (Haake Mini Lab II, Thermo Scientific, ANTISEL, S.A., Athens, Greece) operated at 140 °C and 100 rpm for 3 min [21]. LDPE pellets were mixed with pure AC and the previously prepared powder of the TO@AC nanohybrid to achieve nominal AC and TO@AC weight percentages of 5, 10, and 15% (see Figure 2). The resultant samples were labeled as LDPE/5AC, LDPE/10AC, LDPE/15AC, LDPE/5TO@AC, LDPE/10TO@AC, and LDPE/15TO@AC. A pure LDPE sample was also prepared for comparison. By following the extrusion process, the LDPE/AC and LDPE/TO@AC samples, as well as the pure LDPE, were thermomechanical transformed into films (see Figure 2). This transformation was achieved by operating a hydraulic press with heated platens at 110 °Cand a constant pressure of 2 MPa for 2 min [12,43].

### 2.4. Physicochemical Characterization of AC, TO@AC Powders, and LDPE/AC and LDPE/TO@AC Films

The methods used to carry out the physicochemical characterization of AC, TO@AC powders, and LDPE/AC and LDPE/TO@AC films were as described in [38,39,40,41]. To study possible crystallinity changes in the AC during the process of TO@AC nanohybrid development, XRD analyses were performed on both pure AC and TO@AC. The crystal structures of the resulting LDPE/AC and LDPE/TO@AC films, as well as the pure LDPE film, were also examined via XRD analyses. All XRD measurements were conducted using a Bruker XRD D8 Advance diffractometer (Bruker, Analytical Instruments, S.A., Athens, Greece) with a LINXEYE XE high-resolution energy-dispersive detector, under experimental conditions set in accordance with previous reports. The possible interactions between AC and the adsorbed TO molecules were examined with FTIR spectroscopy measurements on pure AC and the modified TO@AC nanohybrid. Additionally, the interactions between AC, TO@AC, and the LDPE matrix were investigated via FTIR spectroscopy measurements on the pure LDPE film and all of the obtained LDPE/AC films and LDPE/TO@AC active films. The FTIR measurements employed an FT/IR-6000 JASCO Fourier-transform spectrometer (JASCO, Interlab, S.A., Athens, Greece). Differential scanning calorimetry (DSC) measurements were conducted on pure AC and on the modified TO@AC nanohybrid by using a DSC214 Polyma Differential Scanning Calorimeter (NETZSCH manufacturer, Selb, Germany). The total amount of the TO fraction adsorbed onto AC was determined through thermogravimetric analysis (TGA) experiments performed on pure AC and on the modified TO@AC nanohybrid, utilizing a PerkinElmer Pyris Diamond TGA/DTA instrument (Interlab, S.A., Athens, Greece). Lastly, the surface and cross-sectional morphologies of all resulting LDPE/AC and LDPE/TO@AC films, as well as the pure LDPE film, were scrutinized via SEM analysis with an accompanying EDX analysis. For the SEM analysis, a JEOL JSM-6510 LV SEM Microscope (Ltd., Tokyo, Japan) equipped with an X-Act EDS-detector (Oxford Instruments, Abingdon, Oxfordshire, UK) with an acceleration voltage of 20 kV was employed.

### 2.5. Tensile Measurements of LDPE/AC and LDPE/TO@AC Films

For all the obtained LDPE/AC and LDPE/TO@AC films, as well as for the pure LDPE film, tensile properties were measured according to according to the ASTM D638 method using a Simantzü AX-G 5kNt instrument (Simantzu. Asteriadis, S.A., Athens, Greece), following the methodology described previously in [21,40,44].

### 2.6. Water Vapor Transmission Rate (WVTR) and Water Diffusion Coefficient (D_wv_)

The WVTR values for all of the obtained LDPE/AC and LDPE/TO@AC films, as well as for the pure LDPE film, were measured with a handmade apparatus according to the ASTM E96/E 96M-05 method [43,45]. From the WVTR values, the water vapor diffusion coefficient values (D_wv_) were calculated according to the methodology described in detail recently [44,46]. 

### 2.7. Oxygen Transmission Rate Values (O.T.R.) and Oxygen Diffusion Coefficient (P_O2_)

For all of the obtained LDPE/AC and LDPE/TO@AC films, as well as for the pure LDPE film, O.T.R. values were measured by using an oxygen permeation analyzer (O.P.A., 8001, Systech Illinois Instruments Co., Johnsburg, IL, USA) according to the ASTM D 3985 method (23 °C and 0% RH). From these values, oxygen diffusion coefficient (P_O2_) values were obtained by following the methodology described in detail recently [44,46].

### 2.8. Total Antioxidant Activity of Films

The antioxidant activity of all obtained films was measured using the 2,2-diphenyl-1-picrylhydrazyl radical (DPPH) assay, as described by Brand-Williams et al. [47] and modified recently [48,49]. The DPPH assay method has the advantages of a cheap, simple, and trustworthy method but also exhibits some limitations. The evaluation of the antioxidant capacity of a material by the change in DPPH^•^ UV–vis absorbance at a wavelength of 517 nm must be carefully interpreted, since the light absorbance of the DPPH^•^ at this wavelength may be diminished because of the influence of some other factors, e.g., the reaction of the DPPH^•^ with the analyzed sample, pH, O_2_, light, type of solvent, etc. This is why, as mentioned, the method added a buffer solution in all samples to keep the pH stable. Also, dark-colored and hermetically sealed vials were used to avoid light sensitivity and oxidation processes caused by O_2_. In 4 mL of 30 ppm DPPH ethanolic solution, approximately 5 mg of LDPE/5TO@AC, LDPE/10TO@AC, and LDPE/15TO@AC film was added, and the absorbance at 517 nm was recorded as a function of time for 1 h (60 min), 2 h (120 min), 3 h (180 min), 1 day (1440 min), and 2 days (2880 min). As a blank sample, the absorbance of 4 mL of 30ppm DPPH ethanolic solution without the addition of any film was also recorded as a function of time. The % antioxidant activity was calculated by using Equation (1):% film antioxidant activity = [(A_t,film_ − A_t,blank_)/A_0_] × 100(1)
where A_t,film_ is the absorbance of each film as a function of time, A_t,blank_ is the absorbance of the blank sample as a function of time, and A_0_ is the initial absorbance of each sample solution.

### 2.9. TO Release Test—Calculation of Released TO wt.% Content and TO’s Released Diffusion Coefficient (D_TO_)

The experiments of TO’s release from all of the obtained LDPE/TO@AC active films were conducted using an AXIS AS-60 moisture analyzer (AXIS Sp. z o.o. ul. Kartuska 375b, 80-125 Gdańsk) according to the methodology described previously [41]. Approximately 300 to 500 mg of each film was used for the controlled release experiments. The diffusion coefficient (D_TO_) for TO’s release from the obtained LDPE/xTO@AC films was calculated according to Equation (2):(2)$$\frac{m_{t}}{m_{\infty }}=\sqrt{4\frac{D.t}{\pi .l^{2}}}$$
where *m_t_* and *m_∞_* are the amount of TO released form the film after time *t* and after the equilibrium time *t_eq_*→∞, respectively, *D* is the diffusion coefficient, and *l* is the average film thickness.

The linearization of Equation (2) leads to the slightly modified Equation (3):(3)$${\left(\frac{m_{t}}{m_{\infty }}\right)}^{2}=4\frac{D\times t}{\pi \times \ell^{2}}$$

By employing the pseudo-second-order sorption mechanism model [42], we calculated the desorption rate constant *k*_2_ and the maximum TO desorbed *q_e_* according to the following equation: (4)$$q_{t}=\frac{q_{e}^{2}\times k_{2}\times t}{q_{e}\times k_{2}\times t+1}$$
where *q_t_* = *m_t_*/*m*_0_ and *q_e_* = 1.

### 2.10. Packaging Preservation Test of “Skaloppini Type” Fresh Pork Meat Fillets

The pork fillets were wrapped aseptically between two LDPE@15TO@AC active films with a diameter of 11 cm (see Figure 3). For comparison, pork fillets were also wrapped with two LDPE/15AC films and two pure LDPE films (control samples) (see Figure 3). All of the as-wrapped pork fillets were folded inside the packaging paper of the local meat processing plant Aifantis, without the inner membrane used by the company for packaging. After packaging, the fillets were placed in a preservation chamber at 4 ± 1 °C, and then the lipid oxidation, the heme iron content, the total variable counts (TVCs), and the sensory analysis were measured. 

### 2.11. Packaging Test of Fresh Pork Fillets

#### 2.11.1. Lipid Oxidation of Pork Fillets with Thiobarbituric-Acid-Reactive Substances

The thiobarbituric-acid-reactive substances (TBARS) values of the packed pork fillets after 2, 4, 6, 8, 10, and 12 days were determined using the method of Tarladgis et al. [50], as recently modified by Karabagias et al. [51]. The methodology for determination of the TBARS values of packaged fresh pork fillets was as described in detail recently [40,41].

#### 2.11.2. Heme Iron Content

The heme iron content of the packaged fresh pork fillets was determined according to the method reported by Clark et al. [52], as described in detail recently [40,41]. Heme iron content analyses were carried out every 2 days up to 12 days of storage. 

#### 2.11.3. Total Variable Counts (TVCs) of Pork Fillets

The TVCs were monitored every 2 days up to 8 days of storage at 4 ± 1 °C. Ten grams of pork fillet was removed aseptically from each packaging treatment using a spoon, transferred to a stomacher bag (Seward Medical, Worthing, West Sussex, UK) containing 90 mL of sterile buffered peptone water (ΒPW, NCM0015A, Heywood, BL97JJ, UK) (0.1 g/100 mL of distilled water), and homogenized using a stomacher (LAB Blender 400, Seward Medical, UK) for 90 s at room temperature. For the microbial enumeration, 0.1 mL of serial dilutions (1:10 diluents, buffered peptone water) of pork meat homogenates was spread on the surface of plate count agar (PCA, NCM0010A, Heywood UK). The TVCs were determined every 2 days up to 8 days of storage at 4 ± 1 °C after incubation of each plate for 2 days at 30 °C [53].

#### 2.11.4. Sensory Analysis Testing of Pork Fillets

The sensory properties of pork fillets were scaled from 0 (for the least liked sample) to 5 (most liked sample) points by seven experienced members of the Food Science and Technology Department. At each sampling day, color, odor, cohesion, and taste were evaluated. The test samples were cooked, cut to 1.50 cm × 1.50 cm, and then served at 60 °C. Palate cleansers—room-temperature distilled water and unsalted crackers—were provided between samples. The panelists did not taste those samples that exceeded the limit of acceptability for TVCs (higher than 7 log CFU/g) [54].

### 2.12. Statistical Analysis 

In this study, an extensive array of tests was conducted on a minimum of three separate samples. These tests encompassed an assessment of structural, mechanical, and barrier properties, alongside a range of others, including antioxidant activities, controlled release, TBARS, heme iron, and TVC tests. The results presented in the tables throughout this work are mean values derived from these measurements, with each table also displaying the standard deviation. Equal-means hypothesis testing was performed to evaluate the equality of the means. IBM’s SPSS Statistics software, version 20, was used to perform the necessary statistical analyses. Hypothesis tests were carried out via the ANOVA method, using the Tukey criteria for equal-variance assumptions, setting confidence intervals at 95% and significance levels at *p* < 0.05. In addition, the assurance of mean equality (EA%) or inequality (IA%) was examined based on empirical equations previously reported in the literature [55]. Across all tests, it was observed that the mean values of the various sample properties that were statistically different exhibited an inequality assurance factor (IA%) of over 56%, while the mean values that were statistically equal exhibited an equality assurance factor (EA%) of over 61%. More details about the ANOVA results are available in the relevant tables.

## 3. Results and Discussion

### 3.1. Physicochemical Characterization of AC and TO@AC Nanohybrid

The XRD plot of pure AC powder (depicted as line (1) in Figure 4a) is consistent with a typical amorphous material. After the adsorption of TO onto AC to develop the TO@AC nanohybrid, a broad peak appears at around 18° 2theta in the XRD plot of the TO@AC nanohybrid (see line (2) in Figure 4a), suggesting that the adsorption of TO has subtly influenced the amorphous structure of pure AC. Line (1) in the FTIR graph (Figure 4b) corresponds to the FTIR plot of pure thyme oil. The bands observable at around 3530 to 3433 cm^−1^ are attributed to the stretching vibration of O-H groups. The bands at around 3100–3000 cm^−1^ are ascribed to the stretching vibrations of aromatic and alkenic groups of TO molecules. The bands at 2958 and at 2868 cm^−1^ are assigned to the stretching mode of C-H groups. The bands between 1500 cm^−1^ and 1300 cm^−1^ are assigned to the bending of C-H on the C-O-H group and the bending of aliphatic CH_2_ groups [38,39,40]. The FTIR spectra of pure AC (see line (2) in Figure 4b) prominently display bands at 1590 cm^−1^ and 1260 cm^−1^. The band at 1590 cm^−1^ corresponds to the stretching vibration modes of carbonylic (-COO-) groups, while the band at 1260 cm^−1^ is assigned to the stretching vibration of C-O groups [42]. The band with a maximum at approximately 3420–3440 cm^−1^ is attributed to the O–H stretching mode of hydroxyl groups of the adsorbed water molecules [56]. In the FTIR plot of the modified TO@AC nanohybrid (line (3) in Figure 4b), it is evident that in addition to the bands of the pure AC, the characteristic bands of TO also exist in the ranges of 2800–3100 cm^−1^, 1300–1500 cm^−1^, and 500–1000 cm^−1^ [38,39,40]. This observation signifies the adsorption of TO molecules onto the internal surface area of pure AC, without causing a shift in the characteristic peaks of AC. This suggests that the TO molecules are more likely physiosorbed than chemisorbed on the surface of AC, favoring the physisorption of TO molecules on the AC surface for controlled-release applications of such TO@AC nanohybrids.

The thermogravimetric analysis (TG) plot of pure AC, shown as line (1) in Figure 4c, reveals a mass-loss step below 200 °C, which is consistent with the desorption of AC moisture [57]. The TG plot of AC remains almost unchanged until 600 °C, where a second mass-loss step commences [58]. This step correlates with the decomposition of cellulose, lignin, and hemicellulose matter, along with the expulsion of volatile matter that occurs during the carbonization process [59]. The TG plot of the TO@AC nanohybrid (see line (2) in Figure 4c) displays a first mass-loss step below 300 °C and a second one beginning at around 350–400 °C and ending around 500 °C. This pattern suggests that a fraction of TO desorbs from the AC surface in the temperature range from 150 to 300 °C, while another TO fraction desorbs in a higher temperature range from 350 to 500 °C. Considering the previously studied textural features of the AC [42], the Brunauer–Emmett–Teller surface area (S_gBET_) of the AC was 1372 m^2^/g, with the micropore volume being 84.6% and exhibiting micropores at D_micro1_ = 1.28 and D_micro2_ = 1.6 nm [41]. Given that the size of a TO molecule is roughly equal to that of a phenol molecule (about 0.6–0.8 nm), a fraction of TO could be adsorbed within the micropore structure of AC. Consequently, the TO fraction residing in the micropores of AC requires more energy to desorb (second mass-loss step). It is therefore suggested that the first mass-loss step from 150 to 300 °C corresponds to the desorption of TO molecules adsorbed in the mesoporous AC structure, while the second mass-loss step from 350 to 500 °C corresponds to the TO molecules desorbed from the microporous AC structure. In Figure 3c, the % mass loss of the first step and the % total mass loss are calculated by subtracting the mass values of TO@AC from the mass values of pure AC at 300 °C and 500 °C, respectively. The total TO mass adsorbed was found to be equal to 50.7 wt%—a value considerably higher than those recently obtained for TO’s adsorption on natural zeolite (35.5 wt%) [41] and HNTs (31.4 wt%) [40]. This outcome reveals that AC is a promising bio-based material for application as a nanocarrier in controlled-release applications within the active food packaging sector. The DSC plot of pure AC (see line (1) in Figure 4d) exhibits an exothermic peak at approximately 157 °C, corresponding to the enthalpy of water molecules’ desorption. The DSC plot of the modified TO@AC displays a peak at around 217 °C. This peak aligns more closely with the boiling point of TO (232 °C) than the boiling point of limonene (176 °C), indicating that the fraction of molecules adsorbed in AC is rich in TO molecules, which is consistent with previous publications [40,41].

### 3.2. Physicochemical Characterization of the Obtained LDPE/AC and LDPE/TO@AC Films

#### 3.2.1. XRD Analysis

In Figure 5a, the XRD plots of all LDPE/AC and LDPE/TO@AC films, as well as the pure LDPE film, are shown for comparison.

In all plots, the characteristic peaks of LDPE’s crystal phase at Bragg angles of 2θ1 = 21.5° and 23.75° are observed. It can also be observed, that with the addition of both AC and TO@AC, LDPE’s peaks decreased. In the case of LDPE/AC, films the decrease in LDPE’s characteristic peaks is higher than in the case of LDPE/TO@AC. This is an indication of higher dispersion of the hydrophobic modified TO@AC hybrid in the LDPE matrix compared with pure AC. 

#### 3.2.2. FTIR Spectroscopy 

In Figure 5b, the FTIR plots of all obtained LDPE/AC and LDPE/TO@AC films, as well as the pure LDPE film, are shown for comparison. In all cases, the characteristic peaks of LDPE are obtained. More specifically, the bands at 1460 and 715 cm^−1^ are assigned to the asymmetric stretching of the CH_3_ group, the group wagging of the CH_2_ group, and the group rocking of the CH_2_ group of the LDPE. In both LDPE/AC and LDPE/TO@AC films, the characteristic peaks of LDPE are decreased and the characteristic peak of AC at 1590 cm^−1^ is obtained, along with the broad peaks of AC in the range of 900–1260 cm^−1^. No shift in LDPE’s characteristic peaks is observed, implying no chemical bonding between the LDPE matrix and AC or TO@AC’s chemical groups [60]. With a more careful glance, it can be observed that the characteristic peak of AC at 1590 cm^−1^ is more prominent in the case of all LDPE/AC films than in the case of LDPE/TO@AC films. This could be an indication of higher dispersion of the modified and more hydrophobic TO@AC nanohybrid in the LDPE matrix compared with the relevant of pure AC, which is in accordance with the aforementioned XRD observations. Finally, no TO peaks are observed in the case of TO@AC-containing films, and this is an indication that TO molecules are blended inside the LDPE polymer matrix.

#### 3.2.3. SEM Images 

The surface/cross-section morphologies of the pure polymer matrix LDPE, pure AC, and all of the obtained LDPE/xAC and LDPE/xTO@AC films were investigated using an SEM instrument, and the results confirmed that the AC and the hybrid TO@AC nanostructure were homogeneously dispersed in the polymer matrix. The SEM images (surface and cross-section) in Figure 6a,b exhibit the expected homogeneous structure of the pristine polymer matrix. In Figure 6c, the surface morphology of activated carbon (AC) with highly porous characteristics is shown, resembling a honeycomb structure. The micro- and mesoporous surface morphology in AC acts as an absorbent from the packaged food or as a nanocarrier for bioactive compounds released in the packaged food [61].

Surface and relative cross-section images of LDPE/xAC and LDPE/xTO@AC with different ratios (5, 10, 15%) of AC and TO@AC are presented in Figure 7, Figure 8 and Figure 9. 

Based on the SEM studies, it should be mentioned that in all cases of LDPE/xAC and LDPE/xTO@AC films, both AC and TO@AC were homogeneously dispersed, indicating their enhanced compatibility and their interfacial adhesion with the polymer matrix of LDPE. This result agreed with the XRD and FTIR results discussed above. It is obvious that pure AC derived from spent coffee has a high dispersibility in the LDPE matrix due to its hydrophobic nature [42]. Its hydrophobic nature is enhanced by the adsorbed TO molecules, further enhancing its compatibility with LDPE. 

### 3.3. Tensile Properties of LDPE/AC and LDPE/TO@AC Films

The calculated elastic modulus (E), ultimate strength (σ_uts_), and elongation at break (%ε) values for all obtained LDPE/AC and LDPE/TO@AC films, as well as for the pure LDPE film, are listed in Table 1 for comparison.

As shown in Table 1, the incorporation of pure AC powder and the modified TO@AC nanohybrid in the LDPE matrix did not significantly affect the ultimate strength values of LDPE/AC and LDPE/TO@AC films as compared to the ultimate strength value of the pure LDPE film. On the other hand, the incorporation of pure AC powder and the modified TO@AC nanohybrid in the LDPE increased the elastic modulus and % elongation at break values of both LDPE/AC and LDPE/TO@AC films. In advance, the highest % elongation at break value was obtained for LDPE/TO@AC, as compared to the % elongation at break value of the LDPE/AC film. The higher % elongation at break values of the LDPE/AC films as compared to the % elongation at break value of the pure LDPE film suggest the high dispersity and compatibility of AC powder in the LDPE matrix. In contrast to our previous reports, where inorganic nanostructures such as HNTs and NZ were incorporated in the LDPE matrix and led to fragile LDPE/HNT and LDPE/NZ films, the AC powder seems to be more hydrophobic, less inorganic and, thus, more compatible with the LDPE matrix [40,41]. Recently, Sadakpipanich et al. [37] showed that the incorporation of MC activated carbon into the LDPE increased the tensile strength and decreased the elongation at break of the obtained LDPE/MC films. These results are in contrast with the results presented here. However, Sadakpipanich et al. [37] have incorporated MC contents lower than 5 wt.%. into LDPE films. Thus, herein, it is reported for first time that the incorporation of high contents of AC (varying from 5 to 15%) in LDPE drives the development of LDPE/AC films with similar or higher elongation at break values in comparison to pure LDPE. It seems that the higher amounts of AC incorporated into LDPE somehow react as plasticizer. In advance, the plasticization capacity of AC increased for TO@AC nanohybrids because of the presence of TO molecules [12,62,63]. Overall, the highest elastic modulus and % elongation at break values were found for the LDPE/10TO@AC film. The addition of AC or TO@AC in the LDPE matrix imposes a limit. Above this amount, a worse dispersion and homogeneity start to exist. This is why the mechanical properties of the LDPE/15TO@AC became worse in comparison to the relevant properties of the LDPE/10TO@AC film. So, this film had 98.3% and 158.7% higher elastic modulus, and % elongation at break values, respectively, than the pure LDPE film. The abovementioned conclusions were tested via the ANOVA statistical tool, and the results showed that for a significance level of *p* = 0.05 the mean value of the Young’s modulus of LDPE/5AC was statistically equal to the relevant value of the LDPE/5TO@AC. Moreover, the mean value of Young’s modulus of the LDPE/5TO@AC film was statistically equal to the respective value of the LDPE/15TO@AC film, but not equal to the relevant value of the LDPE/5AC film. Equality of the mean values of the elongation at break property was also observed for the LDPE/10AC and LDPE/15TO@AC films. Finally, more details for the statistical procedure comparing the mean values are available in Section 2.12.

### 3.4. Water and Oxygen Barrier Properties of the LDPE/AC and LDPE/TO@AC Films

In Table 2, the obtained water vapor transmission rate (WVTR) and oxygen transmission rate (OTR) values of all LDPE/AC and LDPE/TO@AC films, as well as for the pure LDPE film, are listed. From these values, the water vapor diffusion coefficient (D_wv_) values and the oxygen permeability (Pe_O2_) values for all LDPE/AC and LDPE/TO@AC films, as well as for the pure LDPE film, were calculated, as listed in Table 2 for comparison. 

As shown in Table 2, both the AC powder and the TO@AC nanohybrid successfully increased both the water and oxygen barriers. The higher the AC and TO@AC wt.% contents, the stronger the obtained water and oxygen barriers. A higher increase in the water and oxygen barriers was observed for the TO@AC-containing films, in accordance with the higher dispersion of LDPE/TO@AC as compared to LDPE/AC films, as shown in the SEM images section above. So, the lowest water vapor diffusion coefficient (D_wv_) and oxygen permeability (Pe_O2_) values were obtained for the LDPE/15TO@AC film. For this film, the obtained water vapor diffusion coefficient (D_wv_) value was 55% lower than the water vapor diffusion coefficient (D_wv_) value of the pure LDPE film, while its oxygen permeability (Pe_O2_) value was 95.4% lower than the oxygen permeability (Pe_O2_) value of the pure LDPE film. Sadakpipanich et al. [37] developed LDPE-based films by incorporating AC produced from biomass derived from macadamia nut cultivation. Their findings showed lower water and oxygen barrier properties by varying the AC’s wt%. content from 1 to 5%. Therefore, herein, we have shown for the first time that AC derived from spent coffee is a promising material to be applied as a nanoreinforcement to reduce water/oxygen permeability in active packaging films when incorporated into LDPE films in high wt% contents, from 5 to 15%. Its capacity to increase the water/oxygen barrier is maximized when modified with TO to obtain TO@AC nanohybrids. The abovementioned conclusions were tested via the ANOVA statistical tool, and the results showed that for a significance level of *p* = 0.05, the water vapor barriers of pure LDPE and LDPE/5AC are equal. Equality of the mean values of this property was also observed for the LDPE/15AC and LDPE/10TO@AC films. Finally, the LDPE/15AC film exhibited the same mean value for oxygen barrier as the LDPE/5TO@AC film. More details for the mean values comparing statistical procedures are available in Section 2.12.

### 3.5. Antioxidant Activity of LDPE/TO@AC Films

The plots of % antioxidant activity of all LDPE/TO@AC films as a function of time are shown in Figure 10. 

As shown in Figure 10, the total antioxidant activity of the films increased rapidly during the first day and then remained constant. As expected, the antioxidant activity increased as the wt%. TO content in the films increased. So, after 1 day of incubation, the LDPE/5TO@AC, LDPE/10TO@AC, and LDPE/15TO@AC films reached a total antioxidant activity of 30.2%, 39.0%, and 44.4%, respectively.

### 3.6. TO Release Test—Calculation of Released TO wt.% Content and TO’s Released Diffusion Coefficient (D_TO_) of LDPE/TO@AC Films

By using the data given in Appendix A and the Equations (2)–(4), the diffusion coefficient of TO released, the total amount of TO released (*q_e_*), and the desorption rate constant value (K_2_) for all studied LDPE/xTO@AC films were calculated, as listed in Table 3 for comparison.

As shown in Table 3, as the wt.% content of TO@AC increased, the value of *q_e_* increased, and the value of the constant K_2_ decreased. This means that by increasing the content of TO@AC nanohybrid in the LDPE/xTO@AC active films, the total amount of TO released increases and its release rate is reduced. On the other hand, the calculated values of the TO diffusion coefficient were 1.39 × 10^−7^ cm^2^/s for the LDPE/5TO@AC film, 2.00 × 10^−7^ cm^2^/s for the LDPE/10TO@AC film, and 1.47 × 10^−7^ cm^2^/s for the LDPE/15TO@AC film. The values of the TO diffusion coefficient and K_2_ release rate constant reported here were an order of magnitude higher than the TO diffusion coefficient values reported recently for the TO diffusion coefficient and K_2_ release rate constant from LDPE/xTO@NZ films [41]. This means that AC releases higher amounts of TO and at higher rates than natural zeolite (NZ) [41]. 

### 3.7. Lipid Oxidation of Pork Fillets

The calculated TBARS values of the low-fat pork fillets wrapped with pure LDPE, LDPE/15AC, and LDPE/15TO@AC films are shown in Table 4 for comparison.

The calculated TBARS values from day 0 to the 12th day were similar to those recently reported in [40,41]. From Table 4, it can be seen that both the LDPE/15AC and LDPE/15TO@AC films were successful in reducing the obtained TBARS values during the 12 days of storage in comparison to the relevant TBARS values of the pure LDPE film. In addition, the lowest TBARS values during the 12 days of storage were obtained for the LDPE/15TO@AC active film. 

So, in accordance with recent reports [40,41], both the LDPE/15AC and LDPE/15TO@AC active films succeeded in significantly decreasing the deterioration of pork fillets due to the lipid oxidation process.

### 3.8. Heme Iron of Pork Fillets

Table 4 also shows the calculated heme iron values. These values, as expected, were decreased over the 12 days of storage for all tested films. The greatest decrease in the heme iron values was obtained for the pork fillets wrapped with the LDPE/15TO@AC active film. For the pork fillets wrapped with the LDPE/15AC active film, the heme iron values were higher than the relevant heme iron values of pork fillets wrapped with the LDPE/15TO@AC film but lower than the relevant heme iron values of pork filets wrapped with the pure LDPE film. Thus, both LDPE/15AC and LDPE/15TO@AC films succeeded in preserving the pork fillets with higher heme iron contents, which is beneficial from a nutritional point of view. In advance, in accordance with previous recent reports, the heme iron contents of pork fillets have a linear correlation with the obtained TBARS values [40,41].

### 3.9. Correlation of TBARS and Heme Iron 

Throughout storage, the bivariate Pearson’s correlation analysis was performed on the obtained TBARS and heme iron content values of the pork fillets. The results of the analysis indicated that significant and positive correlations were observed between the two methods throughout the storage period. The Appendix A provide the corresponding correlations (see Appendix A), which were determined in relation to the packaging treatment and storage time.

### 3.10. Microbiological Changes of Pork Fillets

The TVC gives a quantitative estimate of the population of microorganisms such as bacteria, yeasts, and molds in a food sample capable of forming visible colonies. The microorganisms present in pork fillets, either as a part of their natural microflora or as the result of cross-contamination from other sources, are mostly aerobic microorganisms, and their population is an indicator of the product’s microbiological quality [53]. Table 5 shows the changes in the TVC of pork fillets as a function of the film used and the storage time. 

As shown in Table 5, the initial value of TVC was 3.15 log cfu/g, indicating a very good microbiological quality of the pork meat. According to the ICMSF [64], the upper microbiological limit for acceptable quality of foods’ TVC is 7 log cfu/g. This limit value (7 log cfu/g) was almost reached for pork fillets wrapped with pure LDPE and LDPE/15AC films on the 6th day of storage. For pork fillets wrapped with the LDPE/15TO@AC active film, the TVC limit value (7 log cfu/g) was reached on the 8th day of storage. Thus, it is obvious that the TVC values for pork fillets wrapped with the pure LDPE and LDPE/15AC films were significantly higher (*p* < 0.05) in comparison to the TVC values of pork fillets wrapped with the LDPE/15TO@AC active film. In advance, no statistically significant differences (*p* > 0.05) were observed for pork fillets wrapped with the pure LDPE and LDPE/15AC films. For the pork fillets wrapped with the LDPE/15AC film, we observed significantly lower TVC values from pork fillets wrapped with the pure LDPE film only on the 2nd day of storage. So, it seems that the LDPE/15AC film, which has a higher oxygen barrier than the pure LDPE film, succeeded in preventing microbiological growth in pork fillets during the first 2 days of storage. From day 2 to day 8, the active LDPE/15TO@AC film succeeded in preventing microbiological growth in pork fillets in comparison to the pure LDPE and LDPE/15AC films, due to its ability to release TO in the fillets. Overall, we can conclude that the LDPE/15TO@AC active film succeeded in extending the microbiological shelf-life of pork fillets by 2 days. Considering that both the LDPE/15AC film and the LDPE/15TO@AC film resulted in higher water/oxygen barrier properties than the pure LDPE film, we can conclude that the release of TO is the key factor for the shelf-life extension of wrapped pork fillets.

### 3.11. Sensory Evaluation of Pork Fillets

Sensory properties such as color, odor, taste, and cohesion are major factors for consumers to accept a food product. Off odors in spoiled pork meat could be related to compounds that originate from the growth of certain microorganisms, e.g., dimethyl disulfide, dimethyl sulfide, and propylene sulfide generated by *Pseudomonas* spp. or acetoin, diacetyl, and 3-methylbutanol produced by *homofermentative LAB*, *Enterobacteriaceae*, *or Brochothrix thermosphacta* [54]. Chemical compounds, such as ammonia or amines resulting from protein breakdown, as well as ketones and aldehydes resulting from lipid oxidation, could be responsible for the development of an off odor in meat [54]. The sensory evaluation results of the present study are displayed in Table 6. 

The results showed that the LDPE, LDPE/15AC, and LDPE/15TO@AC films had a similar impact on the color, odor, taste, and cohesion attributes of the pork fillet samples throughout the first 4 days of storage time. At day 6, the pork meat samples packaged in the LDPE, LDPE/15AC, and LDPE/15TO@AC films showed statistically significant differences (*p* < 0.05) in all sensory properties, except for the attribute of odor, which was similar for the LDPE and LDPE/15AC films. Also, on day 6, the samples packaged in LDPE and LDPE/15AC exceeded 7 log CFU/g. On the other hand, the pork meat samples packaged in LDPE/15TO@AC, although no statistically significant differences (*p* < 0.05) were observed in the first 4 days for the sensory analysis parameters (Table 6), had a characteristic and pleasant mild spicy flavor throughout the 6 days of storage according to the panelists’ scores, whereas the TVC did not exceed 7 logCFU/g given the antimicrobial properties of thyme essential oil [51]. This finding is probably related to the low fat content of the ‘’scaloppini’’ type pork fillets studied and the impact of TO [40]. In addition, this packaging technology may be beneficial for the meat industry in terms of ‘’marinated’’ pork meat under active packaging.

## 4. Conclusions

In the current study, the preparation and characterization of a novel TO@AC nanohybrid, the development and characterization of innovative LDPE/xTO@AC active packaging films, and their application for the preservation of pork fillets were studied. It is obvious from the results that the hydrophobic AC derived from spent coffee proved to be an excellent nanocarrier for essential oil components such as TO, yielding a TO@AC nanohybrid with a loading capacity of 100% (g_TO_/g_AC_). This high quantity of the physiosorbed TO molecules in the AC pore structure resulted in LDPE/xTO@AC films with TO release rates higher than the TO release rates recently reported for similar LDPE/xTO@NZ active films where TO molecules were adsorbed on a natural zeolite (NZ) nanofiller. XRD, FTIR results, and SEM images demonstrated that both pure AC and TO@AC materials exhibited excellent dispersity and compatibility with the LDPE matrix. The high dispersity of both pure AC and the TO@AC nanohybrid in LDPE resulted in LDPE/xAC and LDPE/xTO@AC films having superior mechanical properties compared to the relevant properties of the pure LDPE film. However, it seems that an upper limit to the weight concentration for the TO@AC material in the LDPE matrix exists and is equal to 10%. The additional loading of such material to 15% started to cause inhomogeneities and the mechanical properties started to become worse compared with the relevant properties of the pure LDPE. 

Nevertheless, because such mechanical behavior still remains far improved compared with the behavior of the pure LDPE film, it makes sense to characterize the LDPE/15TO@AC film as the most promising candidate for food preservation (Young’s modulus 1.5 times higher and elongation at break 2 times higher than the relevant values of pure LDPE), because the water/vapor diffusion coefficient is 230% lower than that of pure LDPE, the oxygen permeability coefficient is 1928% lower than that of pure LDPE, the TO controlled release rate constant is the lowest (k_2_ = 1.488 s^−1^), the antioxidant activity according to the DPPH assay is the highest (44.4%), and the antibacterial activity according to TVC is the best.

All of the abovementioned parameters strongly suggest that the LDPE/15TO@AC active film could potentially extend the shelf-life of pork fillets by at least 2 days.

## Figures and Tables

**Figure 1 foods-12-02590-f001:**
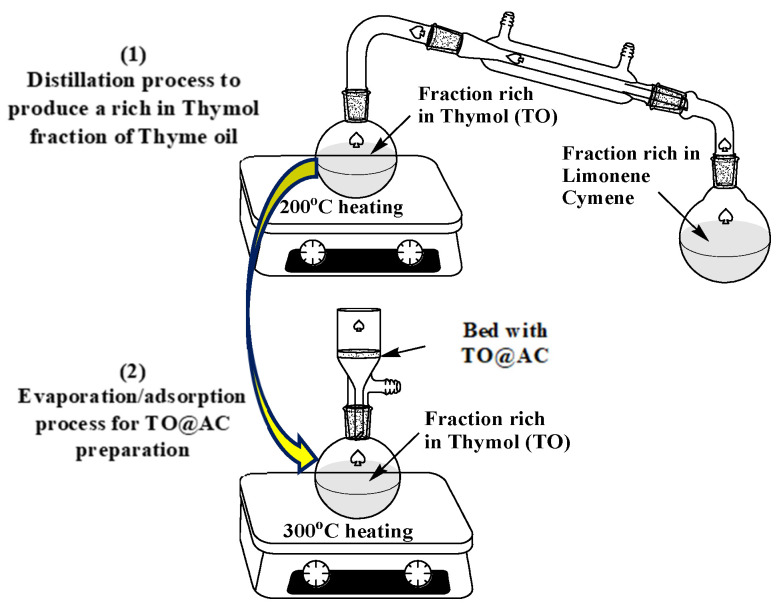
Schematic presentation of the process used for the TO@AC nanohybrid’s preparation.

**Figure 2 foods-12-02590-f002:**
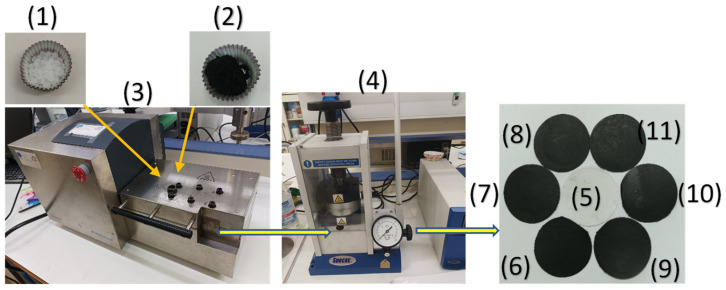
Schematic representation of the extrusion process for the preparation of LDPE/AC and LDPE/TO@AC films: (1) LDPE pellets, (2) TO@AC powder, (3) Mini Lab twin-screw extruder, (4) lab hydraulic press with heated platens, (5) LDPE film, (6) LDPE/5AC film, (7) LDPE/10AC film, (8) LDPE/15AC film, (9) LDPE/5TO@AC film, (10) LDPE/10TO@AC film, and (11) LDPE/15TO@AC film.

**Figure 3 foods-12-02590-f003:**
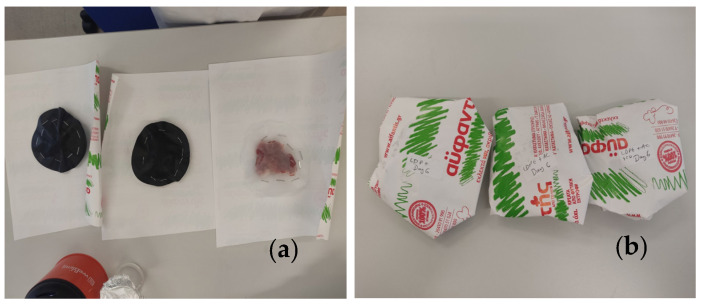
(**a**) Wrapping of fresh pork fillets inside two LDPE@15TO@AC, LDPE/15AC, and pure LDPE films; (**b**) folding of wrapped pork fillets inside the packaging paper of the Aifantis company.

**Figure 4 foods-12-02590-f004:**
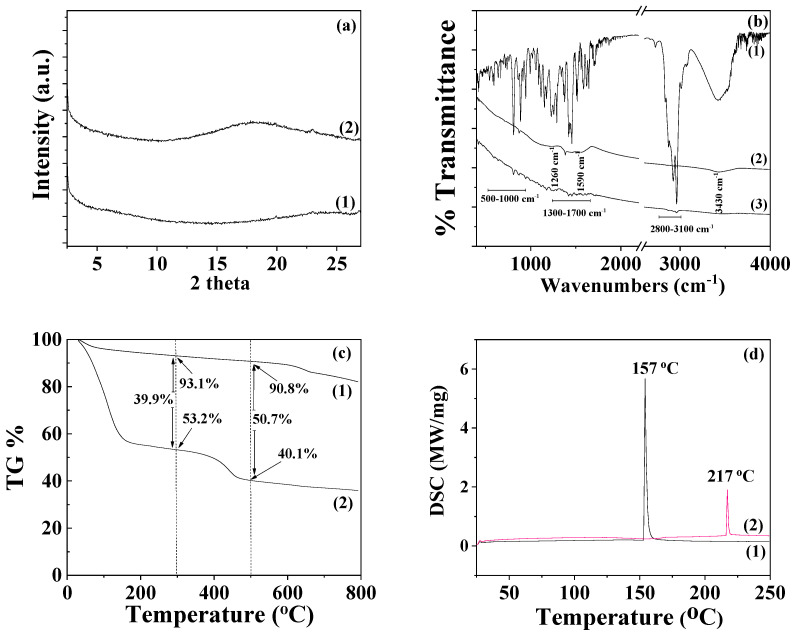
(**a**) X-ray diffraction (XRD) plots of (1) pure AC and (2) modified TO@AC nanohybrid in the range of 2–30° 2 theta. (**b**) Fourier-transform infrared (FTIR) plots of (1) pure thyme oil, (2) pure AC, and (3) modified TO@AC nanohybrid in the range of 400–4000 cm^−1^. (**c**) Thermogravimetric analysis (TGA) plots of (1) pure AC and (2) modified TO@AC nanohybrid in the temperature range of 25 to 800 °C. (**d**) Differential scanning calorimetry (DSC) plots of (1) pure AC and (2) modified TO@AC nanohybrid in the temperature range of 0 to 250 °C. AC: activated carbon, TO: thymol.

**Figure 5 foods-12-02590-f005:**
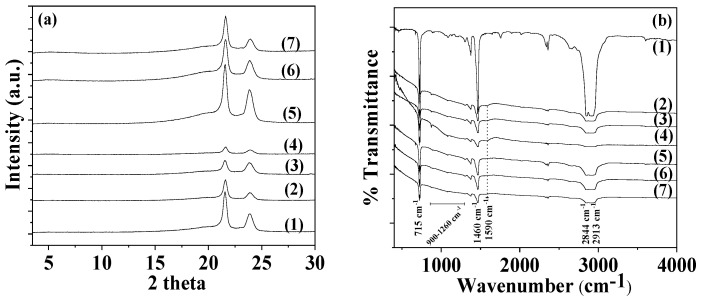
(**a**) X-ray diffraction (XRD) plots of pure LDPE and all LDPE/AC and LDPE/TO@AC films. (**b**) Fourier-transform infrared (FTIR) plots of pure LDPE and all LDPE/AC and LDPE/TO@AC films. LDPE: low-density polyethylene, AC: activated carbon, TO: thymol.

**Figure 6 foods-12-02590-f006:**
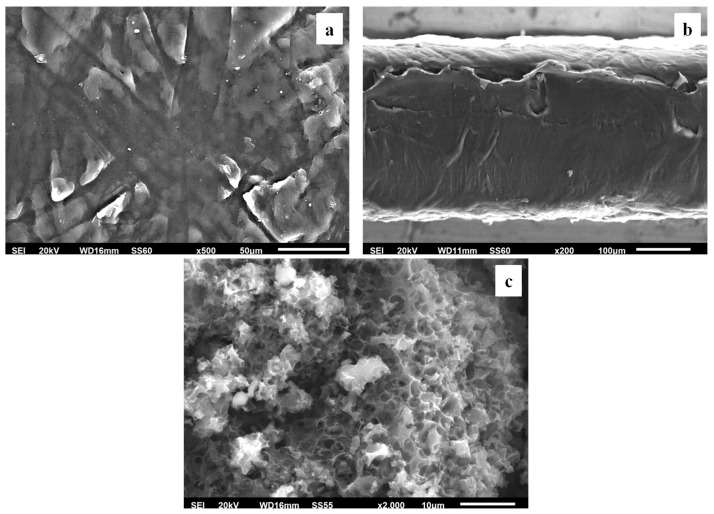
Scanning electron microscopy (SEM) images of the (**a**) surface and (**b**) cross-section for the pure LDPE film, and (**c**) surface morphology of AC. LDPE: low-density polyethylene, AC: activated carbon.

**Figure 7 foods-12-02590-f007:**
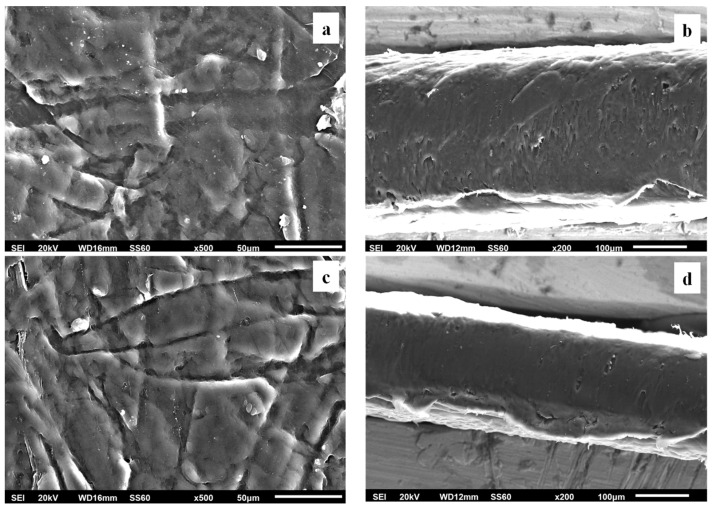
Scanning electron microscopy (SEM) images of the surface (**a**,**c**) and cross-section (**b**,**d**) of the nanocomposite films of (**a**,**b**) LDPE/5AC and (**c**,**d**) LDPE/5TO@AC. LDPE: low-density polyethylene, AC: activated carbon, TO: thymol.

**Figure 8 foods-12-02590-f008:**
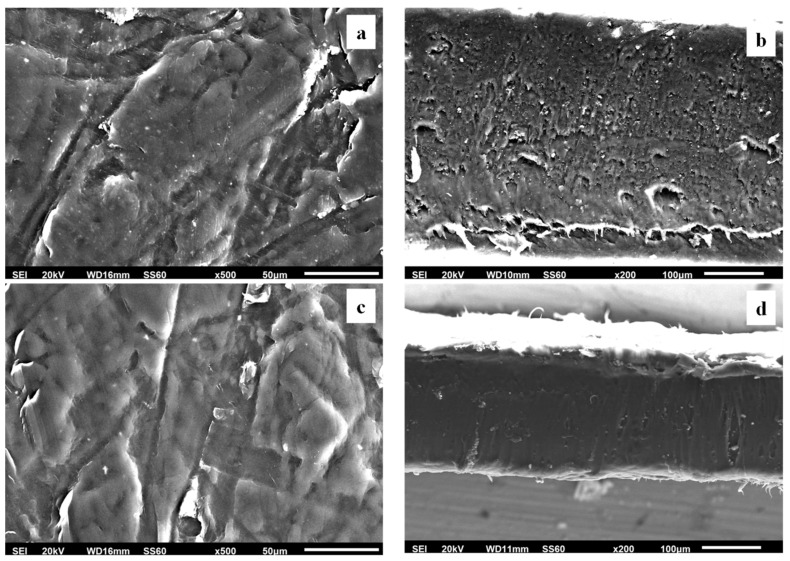
Scanning electron microscopy (SEM) images of the (**a**,**c**) surface and (**b**,**d**) cross-section of the nanocomposite films of (**a**,**b**) LDPE/10AC and (**c**,**d**) LDPE/10TO@AC. LDPE: low-density polyethylene, AC: activated carbon, TO: thymol.

**Figure 9 foods-12-02590-f009:**
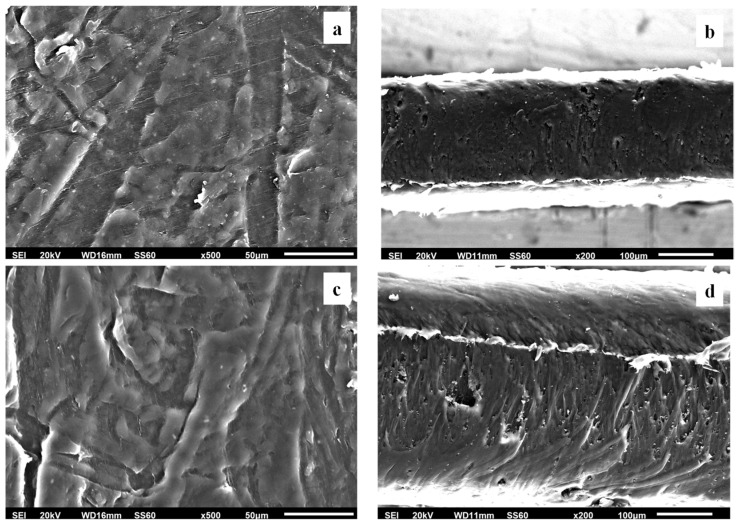
Scanning electron microscopy (SEM) images of the (**a**,**c**) surface and (**b**,**d**) cross-section of the nanocomposite films of (**a**,**b**) LDPE/15AC and (**c**,**d**) LDPE/15TO@AC. LDPE: low-density polyethylene, AC: activated carbon, TO: thymol.

**Figure 10 foods-12-02590-f010:**
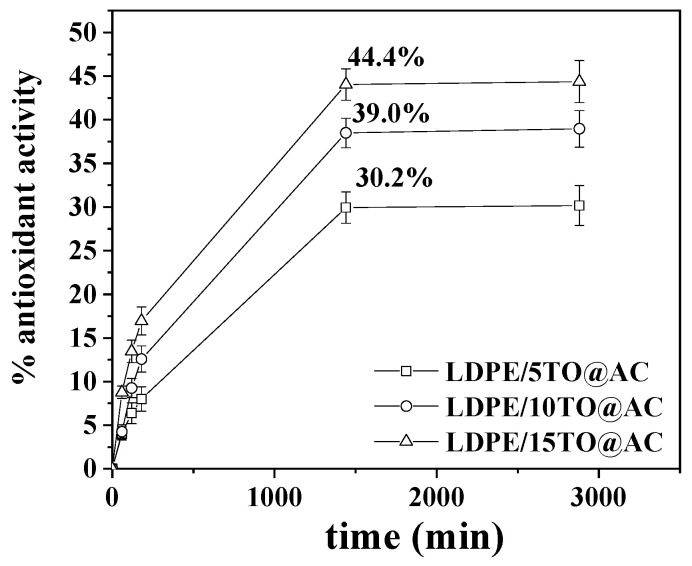
Plots of calculated % antioxidant activity of LDPE/5TO@AC, LDPE/10TO@AC, and LDPE/15TO@AC films as a function of time. DPPH assay method as described by Brand-Williams et al. [47] and modified recently [48,49]. LDPE: low-density polyethylene, AC: activated carbon, TO: thymol.

**Table 1 foods-12-02590-t001:** Elastic modulus (E), ultimate strength (σ_uts_), and elongation at break (%ε) values for all obtained LDPE/AC and LDPE/TO@AC films, as well as for pure LDPE film. LDPE: low-density polyethylene, AC: activated carbon, TO: thymol.

	E (MPa)	σ_uts_ (MPa)	%ε
LDPE	183.3 ± 28.8	12.6 ± 0.5	29.3 ± 8.1
LDPE/5AC	282.7 ± 17.1 ^a^	10.9 ± 1.5	33.4 ± 2.7
LDPE/10AC	312.7 ± 21.4 ^b^	11.8 ± 1.9	63.4 ± 14.0 ^d^
LDPE/15AC	327.3 ± 21.8 ^b^	12.7 ± 1.9	51.0 ± 12.7
LDPE/5TO@AC	286.0 ± 14.5 ^a,c^	11.2 ± 1.7	43.3 ± 17.5
LDPE/10TO@AC	363.5 ± 19.4	12.1 ± 2.4	75.8 ± 15.1
LDPE/15TO@AC	291.0 ± 19.0 ^c^	11.8 ± 1.0	64.8 ± 18.6 ^d^

^a,b,c,d^ Indices for statistically equal mean values according to the ANOVA comparison method and Tukey’s criteria for the assumption of equal variances. Significance level: *p* < 0.05.

**Table 2 foods-12-02590-t002:** WVTR, D_wv_, OTR, and Pe_O2_ values for all LDPE/AC and LDPE/TO@AC films, as well as for the pure LDPE film. LDPE: low-density polyethylene, AC: activated carbon, TO: thymol.

	Film Thickness (mm)	WVTR (10^−7^ gr cm^−2^ s^1^)	D_wv_ (10^−4^ cm^2^/s)	Film Thickness (mm)	OTR (mL m^−2^ day^−1^)	Pe_O2_ (10^−8^ cm^2^/s)
LDPE	0.270 ± 0.012	5.89 ± 0.13	3.77 ± 0.17 ^a^	0.270 ± 0.010	6407 ± 120	20.02 ± 2.41
LDPE/5AC	0.256 ± 0.011	6.42 ± 0.56	3.73 ± 0.07 ^a^	0.266 ± 0.004	1365 ± 62	4.19 ± 0.2
LDPE/10AC	0.141 ± 0.014	7.31 ± 0.20	2.35 ± 0.02	0.378 ± 0.010	818 ± 15	3.57 ± 0.21
LDPE/15AC	0.176 ± 0.010	5.02 ± 0.73	1.99 ± 0.31 ^b^	0.410 ± 0.015	675 ± 34	3.20 ± 0.19 ^c^
LDPE/5TO@AC	0.332 ± 0.012	3.96 ± 0.53	2.84 ± 0.23	0.170 ± 0.015	1593 ± 67	3.12 ± 0.17 ^c^
LDPE/10TO@AC	0.186 ± 0.016	4.55 ± 0.13	1.87 ± 0.21 ^b^	0.311 ± 0.005	642 ± 17	2.11 ± 0.16
LDPE/15TO@AC	0.241 ± 0.013	2.62 ± 0.23	1.47 ± 0.04	0.351 ± 0.010	228 ± 6	0.92 ± 0.04

^a,b,c^ Indices for statistically equal mean values according to the ANOVA comparison method and Tukey’s criteria for the assumption of equal variances. Significance level: *p* < 0.05.

**Table 3 foods-12-02590-t003:** Calculated values of the diffusion coefficient of TO molecule, total desorbed amount of TO (*q_e_*), and desorption rate constant (K_2_) for all obtained LDPE/xTO@AC active films. LDPE: low-density polyethylene, AC: activated carbon, TO: thymol.

	D_TO_ × 10^−7^ (cm^2^/s)	*q_e_*	K_2_ (s^−1^)
LDPE/5TO@AC	1.39 ± 0.45	0.023 ± 0.002	2.425 ± 0.419
LDPE/10TO@AC	2.00 ± 0.33	0.032 ± 0.006	2.444 ± 0.355
LDPE/15@TOAC	1.47 ± 0.86	0.044 ± 0.009	1.488 ± 0.142

**Table 4 foods-12-02590-t004:** Calculated TBARS and heme iron content values of pork fillets wrapped with pure LDPE, LDPE/15AC, and LDPE/15TOAC films, with respect to storage time. LDPE: low-density polyethylene, AC: activated carbon, TO: thymol.

TBARS	Day 0	Day 2	Day 4	Day 6	Day 8	Day 10	Day 12
	AVG ± SD						
	(mg/kg)						
Control	0.23 ^a^ ± 0.01	0.33 ^b^ ± 0.02	0.46 ^d^ ± 0.01	0.63 ^g^ ± 0.03	0.98 ^j^ ± 0.02	1.21 ^m^ ± 0.02	1.36 ^p^ ± 0.02
LDPE/15AC	-	0.28 ^b^ ± 0.01	0.40 ^e^ ± 0.01	0.53 ^h^ ± 0.01	0.83 ^k^ ± 0.03	1.10 ^n^ ± 0.04	1.28 ^q^ ± 0.01
LDPE/15TO@AC	-	0.23 ^c^ ± 0.01	0.33 ^f^ ± 0.02	0.44 ^i^ ± 0.02	0.75 ^l^ ± 0.03	0.95 ^o^ ± 0.03	1.13 ^r^ ± 0.02
	**ANOVA**						
F	-	39.577	63.033	82.765	91.031	67.336	149.763
*p*	-	0.000	0.000	0.000	0.000	0.000	0.000
**Fe**	**Day 0**	**Day 2**	**Day 4**	**Day 6**	**Day 8**	**Day 10**	**Day 12**
	**AVG ± SD**						
	**(μg/g)**						
Control	10.18 ^a^ ± 0.12	8.00 ^b^ ± 0.12	6.16 ^e^ ± 0.12	5.12 ^h^ ± 0.19	3.54 ^k^ ± 0.16	2.22 ^n^ ± 0.27	1.28 ^p^ ± 0.15
LDPE/15AC	-	8.96 ^c^ ± 0.09	6.80 ^f^ ± 0.09	5.72 ^i^ ± 0.12	4.14 ^l^ ± 0.18	2.48 ^n^ ± 0.15	1.82 ^q^ ± 0.09
LDPE/15TO@AC	-	9.66 ^d^ ± 0.12	7.92 ^g^ ± 0.12	7.40 ^j^ ± 0.27	5.70 ^m^ ± 0.16	3.84 ^o^ ± 0.18	2.48 ^r^ ± 0.15
	**ANOVA**						
F	-	162.781	186.000	99.771	135.130	52.083	60.200
*p*	-	0.000	0.000	0.000	0.000	0.000	0.000

Different letters in the same column indicate statistically significant differences at the confidence level *p* < 0.05 (see also Appendix A).

**Table 5 foods-12-02590-t005:** Calculated TVC values of pork fillets wrapped with pure LDPE, LDPE/15AC, and LDPE/15TOAC with respect to storage time. LDPE: low-density polyethylene, AC: activated carbon, TO: thymol. PCA used: (PCA, NCM0010A, Heywood UK). The plates seeded by the spread plate method were incubated for 2 days at 30 °C.

Days					
Sample Name	0	2	4	6	8
	logCFU/g (Avg ± SD)				
LDPE	3.15 ± 0.06 ^a^	4.56 ± 0.32 ^b^	5.65 ± 0.24 ^d^	6.98 ± 0.41 ^f^	7.88 ± 0.20 ^h^
LDPE/15AC	3.15 ± 0.06 ^a^	4.15 ± 0.12 ^c^	5.52 ± 0.18 ^d^	6.68 ± 0.32 ^f^	7.53 ± 0.21 ^h^
LDPE/15TO@AC	3.15 ± 0.06 ^a^	3.86 ± 0.09 ^c^	4.53 ± 0.08 ^e^	5.77 ± 0.17 ^g^	6.84 ± 0.14 ^i^
ANOVA					
F *	-	8.914	35.032	11.935	28.478
P *	-	0.0160	0.0000	0.0080	0.0010

Identical letters in each column indicate no significant statistical differences. * ANOVA results, confidence level *p* < 0.05 (see also Appendix A).

**Table 6 foods-12-02590-t006:** Taste and odor of wrapped pork fillets during storage at 4 ± 1 °C.

Taste					Odor			
Sample Name	0 Day	2nd Day	4th Day	6th Day	0 Day	2nd Day	4th Day	6th Day
LDPE	5.00 ± 0.00 ^a^	4.70 ± 0.29 ^b^	4.46 ± 0.49 ^c^	-	5.00 ± 0.00 ^e^	4.70 ± 0.19 ^f^	4.16 ± 0.51 ^g^	3.22 ± 0.36 ^h^
LDPE/15AC	5.00 ± 0.00 ^a^	4.68 ± 0.22 ^b^	4.42 ± 0.35 ^c^	-	5.00 ± 0.00 ^e^	4.70 ± 0.41 ^f^	4.16 ± 0.44 ^g^	3.44 ± 0.22 ^h^
LDPE/15TO@AC	5.00 ± 0.00 ^a^	4.88 ± 0.22 ^b^	4.70 ± 0.30 ^c^	4.10 ± 0.16 ^d^	5.00 ± 0.00 ^e^	4.86 ± 0.13 ^f^	4.5 ± 0.31 ^g^	3.92 ± 0.16 ^i^
**Color**					**Cohesion**			
	**0 Day**	**2nd Day**	**4th Day**	**6th Day**	**0 Day**	**2nd Day**	**4th Day**	**6th Day**
LDPE	5.00 ± 0.00 ^j^	4.58 ± 0.39 ^k^	4.08 ± 0.57 ^l^	2.96 ± 0.15 ^m^	5.00 ± 0.00 ^p^	4.48 ± 0.29 ^q^	3.78 ± 0.30 ^r^	2.84 ± 0.11 ^s^
LDPE/15AC	5.00 ± 0.00 ^j^	4.32 ± 0.30 ^k^	4.00 ± 0.53 ^l^	3.20 ± 0.16 ^n^	5.00 ± 0.00 ^p^	4.38 ± 0.40 ^q^	4.12 ± 0.51 ^r^	3.44 ± 0.30 ^t^
LDPE/15TO@AC	5.00 ± 0.00 ^j^	4.58 ± 0.32 ^k^	4.30 ± 0.34 ^l^	4.02 ± 0.19 ^o^	5.00 ± 0.00 ^p^	4.46 ± 0.27 ^q^	4.26 ± 0.43 ^r^	3.94 ± 0.18 ^u^

Identical letters in each column indicate no significant statistical differences (*p* > 0.05) (see also Appendix A).

## Data Availability

The datasets generated for this study are available upon request to the corresponding author.

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
