# Peer review of "Thymol@activated Carbon Nanohybrid for Low-Density Polyethylene-Based Active Packaging Films for Pork Fillets’ Shelf-Life Extension"

_foods, 2023, doi:10.3390/foods12132590_

Round 1
Reviewer 1 Report
In this manuscript, an activated carbon (AC) nanostructure rich in thymol (TO@AC) was prepared and physiochemically characterized. The characterization of the prepared reinforcement is very complete. LDPE films were prepared with the incorporation of different amounts of AC and TO@AC. The effect of AC and TO@AC on the physicochemical properties of the film was studied in deep. In addition, the antioxidant and antibacterial properties of the active films were reported. I found the manuscript very interesting and complete. The methodology described is very solid and the results are very clear and supported by the performed determinations. I suggest some minor changes. Some suggestions are presented in the pdf file.

Author Response
First of all Authors would like to express their appreciation for Reviewers’ effort to improve this work. Attached please find our response.

Reviewer 2 Report
The manuscript FOODS-2460868 presents originality, and the research theme falls within the scope of the Special Issue "Circular Bioeconomy: Novel Processes and Materials for Food Preservation". The manuscript is well founded; however, some adjustments and suggestions are described below, to improve the final version of the manuscript.
INTRODUCTION
Lines 87-91: “The obtained LDPE/MC films... were also studied [36].” The authors describe research that developed LDPE/MC films, and cite the methods used in the study, but the information is loose, without continuity. What is the importance of citing such methodologies? What did the authors of the study with LDPE/MC films find most relevant? Thus, I suggest rewriting the sentence, making clear the importance and relevant results of the research with LDPE/MC films, which was one of the article that encouraged the present authors to carry out the research described in the article FOODS-2460868.
Lines 93-98: “In recent studies,… preserving low-fat pork fillets [39,40].” This information is important, as it is a justification for the choice of some methods used in this article, however there is a lack of continuity between the paragraphs, as well as the reason for the information being listed. Authors should be careful in the written part, and check whether a paragraph continues the next, as well as the information contained in them.
lines 110 – 114 “The obtained TO@AC was... release were studied for all active LDPE/TO@AC films.” It is necessary to report the methods used in the study. Justification of methods used is usually presented in the DISCUSSION. Thus, in my opinion, it is unnecessary to mention the methods in the INTRODUCTION topic. Authors can write the beginning of the paragraph already starting with the objectives of the study.
Line 113: “TBARs” put in full and the abbreviation in parentheses, as this is the first time it appears in the manuscript.
Lines 121-123: “Moreover, the... 2 days for pork fillets.” It is part of the conclusion of the study, so it has no reason to be at the end of the Introduction. I recommend withdrawal.
MATERIALS AND METHODS
Line 146: “HNT and NZ, as described recently [37,38]”, depending on how the INTRODUCTION will be rewritten, I advise removing “as described recently”, just leaving the references [37,38].
2.4 Physicochemical characterization of AC, TO@AC, powders and LDPE/AC and LDPE/TO@AC 170 films: to enable readers to have more detailed access to the conditions under which the characterizations were carried out, place a reference after each method used. If the references to be cited are the same, put an initial sentence, for example: “The methods used to carry out the Physicochemical characterization of AC, TO@AC, powders and LDPE/AC and LDPE/TO@AC films were according to [ xx, yy], put the reference number, I think it will be 36,37,38]. As the authors referenced the other items of the methodology 2.5 , 2.6 .....
2.8. Total Antioxidant activity of films: Also refer to the described method.
2.11.2 Heme iron content - Was Heme iron content also evaluated up to 12 days, or only at the beginning and end? The answer to my question is only answered in Table 5. Therefore, it is important to state in the method that the analyzes were carried out every 2 days, up to 12 days.
2.11.3 Total variable counts (TVC) of pork fillets: The authors describe in detail how the sample will be collected for analysis, the culture medium used, and the spread plate plating method, and the cultivation conditions. However, in my opinion, I think it is relevant to remember/reinforce the storage conditions of pork fillets (temperature and storage time), and the sample collection interval.
Line 270: “with respect to storage time”, the statement I made above can be included here (specify temperature, and storage time and sample collection interval for microbial analysis).
RESULTS AND DISCUSSION
In general, the results are well described, and the discussions are relevant, with explanations based on the results to infer possible mechanisms of interaction between the tested components. As a suggestion, I advise authors to write the name of the techniques in full in the text of the figure captions, and put the abbreviations in parentheses, thus placing at the end of the caption what the acronyms of the substances or films tested mean (AC, TO, TO@AC , LDPE/AC and LDPE/TO@AC). In the tables, the techniques or parameters used are already spelled out, just put the abbreviations of the tested substances or films at the end.
3.5 Antioxidant activity of LDPE/TO@AC films: I would like to know from the authors why they only did one type of test to verify the antioxidant activity? It's just a curiosity, as the DPPH test is one of the most used due to its simplicity and low cost when compared to other tests.
In the caption of Table 10, place the method used to calculate the antioxidant activity.
As the authors opted for only one method, I find it interesting to discuss the advantages and limitations of the DPPH method, to further enrich the discussion of antioxidant activity.
3.7 Lipid Oxidation of pork fillets: Make an initial paragraph, as the authors did in the previous topics, to introduce Table 5. Or just go up the paragraph referring to lines 523-529.
Line 526- “From Table 4 it is obtained… “I believe the authors are referring to Table 5 and not 4. Please, make the change.
3.10 Microbiological changes of pork fillets: When reading the results and discussion of this topic and looking at Table 6, I confess that I was confused. Because earlier in the methodology, the authors, when describing the method, put in lines 277-278 “TVC was determined using after incubation for 2 days at 30 ºC [49].” But Table 6 shows results from time zero to 12 days, with analysis every 2 days apart. It is important to make this correction in the methodology.
Table 6: In the legend, include the culture medium used, plate count agar (PCA), and the culture conditions for the count (for example: the plates seeded by spread plate were incubated for 48 hours at 30ºC).
CONCLUSÕES
Writing conclusions should provide the reader with answers to the research objectives and hypothesis. It should not contain results, but only what the results obtained can answer.
´For example in line 587-589 “In the current study, the preparation and characterization of novel TO@AC nanohybrids, the development and characterization of innovative LDPE/TO@AC active packaging films, and their application in pork fillets preservation were studied in detail” The conclusion can be: In the present study, an unprecedented TO@AC nanohybrids was obtained which, when incorporated into LDPE, made it possible to obtain innovative LDPE/TO@AC active packaging films for use in pork fillets preservation.
Lines 604-606: “In general, LDPE/xTO@AC films exhibited greater elongation-at-break values and superior water/oxygen barrier properties due to the presence of TO molecules. LDPE/xTO@AC active films also showed significant antioxidant activity.” – That's a conclusion!
Author Response

(The authors gave the same response as above.)

Reviewer 3 Report
In this study, the authors developed an active food packaging material using thymol, activated carbon, and LDPE for pork application. However, it is strongly advised that the authors carefully address the following concerns
· · Add quantitative data into abstract
· L41-43 unclear statement! removing food additive?
· l46: EO are not the only natural compound with antimicrobial and antioxidant activities rewrite.
· L48: don not use oil instead of EO
· Define each abbreviate term in the first use.
· Should be clear in the introduction the legal framework of using AC in food products
· Please summarize the last paragraph of the introduction and focus on only objectives and novelty. Novelty should be clearly stated. Why this type of AC was used?
· 147: In this section, the authors employed a technique for creating a thymol-enriched solution. However, this approach appears to be flawed as it results in the presence of additional compounds in the solution, apart from thymol and limonene-cymene. In order to validate this, the authors should conduct an analysis of the thymol-rich solution that was obtained. It is worth noting that the current method exhibits poor repeatability.
· The rationale behind the conditions employed in the production of TO@AC should be explained by the authors. Given the utilization of an unconventional source, it would be valuable to examine the impact of various parameters on both the production process and properties of TO@AC
· the toxicity of all developed materials containing AC should be included since they are in close contact with food.
· Given that both AC and TO compounds can impact the sensory parameters of food, it is essential to conduct sensory analysis on the meat to evaluate their effects. This can affect the practical application of such material in food.
There are many grammatical errors and syntax, please correct any grammatical errors and improve the syntax throughout the manuscript
Author Response

(The authors gave the same response as above.)

Round 2
Reviewer 3 Report
The manuscript still contains grammatical errors and syntax issues, including in the newly added text. It is crucial to thoroughly review the entire manuscript, paying close attention to areas such as the abstract (specifically line 23) and the revised sections. This will help identify and correct any remaining errors, ensuring the manuscript's overall quality and coherence
L122: don not use “”As an overall conclusion”, write sentence in past tense
improve last paragraph of introduction and explain novelty in a good way to express the importance of you work for the readers. also amend sentence grammatical structure to state the objectives.
The authors stated that the sensory parameters of food, influenced by the use of AC and TO, do not present any problems. However, it is important to consider various factors such as the type of food, concentration of the agent, method of use, and more. Therefore, a thorough individual experiment is required to analyze these aspects by sensory member. I do not have knowledge about the specific sensory properties of EO and AC in meat.
The manuscript still contains grammatical errors and syntax issues, including in the newly added text. It is crucial to thoroughly review the entire manuscript, paying close attention to areas such as the abstract (specifically line 23) and the revised sections. This will help identify and correct any remaining errors, ensuring the manuscript's overall quality and coherence
Author Response
Thank ou for your effort to improve our work.
Attached please find our response.
